# Analysis of Clinical Trials and Review of Recent Advances in Therapy Decisions for Locally Advanced Prostate Cancer

**DOI:** 10.3390/jpm13060938

**Published:** 2023-06-01

**Authors:** Norman R. Williams

**Affiliations:** UCL Division of Surgery & Interventional Science, 3rd Floor, Charles Bell House, 43–45 Foley Street, London W1W 7TY, UK; norman.williams@ucl.ac.uk

**Keywords:** male, prostate, neoplasms, X-rays, metal nanoparticles, gold, cell death

## Abstract

Despite the implementation of screening and early detection in many countries, the prostate cancer mortality rate remains high, particularly when the cancer is locally advanced. Targeted therapies with high efficacy and minimal harms should be particularly beneficial in this group, and several new approaches show promise. This article briefly analyses relevant clinical studies listed on ClinicalTrials.gov, combined with a short literature review that considers new therapeutic approaches that can be investigated in future clinical trials. Therapies using gold nanoparticles are of special interest in low-resource settings as they can localize and enhance the cancer-cell killing potential of X-rays using equipment that is already widely available.

## 1. Introduction

Prostate cancer is a disease that affects millions of men [1]. Globally, prostate cancer is the second-most common cancer in men, with an annual incidence of 1.4 million and 375,000 deaths [2]. In spite of the widespread adoption of early detection by prostate-specific antigen (PSA) testing, the age standardized mortality rate has been increasing in the USA, Canada, and some European countries [3], although this might be due to overdiagnosis [4]. There is therefore an increasing need to continue to develop more effective treatments.

Men who die of prostate cancer do so because of metastatic spread from the original tumor, principally to the bone (spine, pelvis, and ribs). These metastatic sites can in turn become secondary sites of metastatic spread, eventually resulting in incurable disease. In a large proportion of these men, it is likely that the disease has already spread by the time of diagnosis and initial treatment [5]. 

Advanced prostate cancer is the term used to describe cases where the tumor has spread beyond the prostate and is usually divided into locally advanced (stage III) cancer, where the spread is confined to nearby tissues within the pelvis, and advanced (stage IV), where the cancer has spread to other parts of the body [6]. Locally advanced spread to the lymph nodes is associated with a poor prognosis, but this varies from 75–80% five-year survival with one lymph node involved, down to 20–30% five-year survival with more than five metastatic lymph nodes [7]. Therefore, the detection and eradication of locally advanced metastatic disease is an important area of research [8].

The ClinicalTrials.gov database is free-to-use and contains details of more than 450,000 privately and publicly funded clinical studies conducted in 221 countries [9]; an analysis of these studies can identify knowledge gaps and emerging trends in research. This article describes the past, ongoing, and planned clinical trials of various therapies for locally advanced prostate cancer, and then describes the results of a literature review of some novel areas of research in this area. This information will be useful to a man with a first diagnosis of locally advanced prostate cancer who has not yet undergone surgical or radiation therapy and wishes to know the current “state of the art” for men in this situation.

## 2. Materials and Methods

Specific search terms were used in the query section of ClinicalTrials.gov. A comma-separated values (CSV) file of all studies meeting the query criteria was downloaded and analyzed using Microsoft^®^ Excel^®^ for Microsoft 365 and JMP^®^ Pro 17.1.0 running on Windows 10. 

For the literature review, PubMed^®^ (comprising more than 35 million citations for the biomedical literature from MEDLINE, life science journals, and online books) was queried, and recent relevant articles were selected primarily based on their relevance to detection and/or treatment of locally advanced and/or lymph node involved prostate cancer. The literature review was not carried out systematically. Most of the information was obtained from review articles and their cited sources. In addition, manufacturers’ websites were scanned to determine if there were clinical trials in the pipeline.

## 3. Results

The results from the ClinicalTrials.gov query and the literature review are considered separately.

### 3.1. Analysis of Studies Reported in ClinicalTrials.gov

A query of [prostate cancer] in “condition or disease” and [lymph node metastases] in “other terms” on 18 May 2023 found 103 studies after removal of 60 studies where the eligibility criteria did not include locally advanced disease (Figure 1). A file listing details of all studies has been included with this article as a Appendix A. Characteristics of these studies are as follows. 

#### 3.1.1. Geographic Distribution

The geographic distribution of the studies is shown in Table 1. Most studies were run in countries where the PI (Principal Investigator) was based in North America (46/103 = 45%), followed by Europe (44/103 = 43%).

#### 3.1.2. Status of Studies

The status of the studies is shown in Figure 2. Most studies had “completed” (the study had ended normally, and the last participant’s last visit had occurred), and a comparable proportion were either “recruiting” or “active and not recruiting”. One study was marked as “suspended” but may start again. The three studies that had been “terminated” had stopped early and will not start again. The two studies marked as “withdrawn” stopped before enrolling the first participant. The 12 studies marked as “unknown” had a last known status of “recruiting”, “not yet recruiting”, or “active and not recruiting” but have passed their completion dates, and the status had not been verified within the past two years.

#### 3.1.3. Start Date

The start dates of the studies are shown in Figure 3 and range from 1990 to 2023. Details of studies with a planned start date in 2022 and 2023 are shown in Table 2.

#### 3.1.4. Phases of Studies

The phases of the studies are shown in Figure 4. For most studies, the phase was “not applicable”, mainly because these were observational or diagnostic studies. 

#### 3.1.5. Study Designs

The types of study designs are shown in Figure 5 and Figure 6. Most studies were non-randomized and either diagnostic or treatment. 

#### 3.1.6. Types of Intervention

Table 3 lists the types of interventions used in the studies. The most frequently reported intervention was “drug” (45 interventions), followed by “procedure” (27 interventions), then “radiation” (22 interventions). Where the intervention was “radiation”, Table 4 lists the types of radiation used; the most frequent was conventional radiation therapy (nine studies). 

#### 3.1.7. Most Recent Studies

Table 2 lists details of the most recent studies (planned start date in 2022 or 2023). Two have a randomized design, four are non-randomized, two are observational, and one study has been withdrawn. 

### 3.2. New Treatments

A literature review of recent advances in the detection and treatment of prostate cancer revealed novel treatments that may not yet have reached the stage where they can be tested in clinical trials. This is not intended to be a systematic overview of the field, rather a brief up-to-date summary with an emphasis on prostate cancer that is locally advanced or confined to the lymphatics.

#### 3.2.1. Radiation Therapy

Radiation therapy is often given as treatment for locally advanced prostate cancer. Areas of research include the combination of radiation therapy with androgen deprivation therapy and dose escalation and hypofractionation aimed at improving the precision and accuracy of radiation delivery in order to improve efficacy and reduce harm [10].

Ongoing clinical trials include PACE-NODES, a phase III randomized trial of five-fraction prostate stereotactic body radiotherapy (SBRT) versus five-fraction prostate and pelvic nodal SBRT [11] and a pilot trial investigating the use of stereotactic radiation therapy (SABR) in the management of lymph node-positive prostate cancer recurrence [12].

#### 3.2.2. Nanoparticle-Enhanced Radiotherapy

Nanoparticle-enhanced radiotherapy (NERT) is a method of enhancing the therapeutic ratio of radiation therapy by the addition of nanoparticles that are located in or near the tumor cells [13]. There are many types in development, including a few that have been tested in clinical studies.

Nano-radiopharmaceuticals

Radiolabeled nanoparticles have the potential to be used both for imaging and therapy, and several are under development. In particular, studies of prostate-specific membrane antigen (PSMA)-targeted radiopharmaceuticals are being conducted [14], including the use of lutetium-177 [^177^Lu]-PSMA-617 in men with metastatic prostate cancer [15,16]. A comprehensive review of this topic has been published recently [17].

Gadolinium-based nanoparticles

AGuIX^®^ (NH TherAguix SA, Meylan, France) is a polysiloxane-based nanoparticle containing approximately 15 gadolinium atoms in a particle of about 5 nanometers in diameter. The small size allows for selective accumulation in tumors after intravenous administration. The gadolinium allows MRI-contrast imaging and enhances the radiation-induced damage to tumors [18]. One first-in-human clinical trial has completed (phase 1b on brain metastases) [19,20] and a further five are in the pipeline (including cervical, pancreatic, and lung cancer and glioblastoma) [21]. To date, there is only pre-clinical evidence from human prostate cancer cell lines (DU145 and PC3) to support a possible role as a radiosensitizer in prostate cancer [22], and therefore there is potential for the use of this compound in a clinical trial of men with locally advanced prostate cancer.

Hafnium oxide particles

NBTXR3 (Nanobiotix, Paris, France) is a radioenhancer composed of hafnium oxide crystalline nanoparticles [23]. Eight clinical trials have completed and a further six are recruiting, including head and neck and esophageal cancer and metastatic disease from lung and pancreas [24]. No clinical trials of NBTXR3 have completed in prostate cancer, although pre-clinical evidence in mice and patient-derived xenograft models is encouraging [25]. A phase 1/2 study of NBTXR3 nanoparticles and radiation therapy in the treatment of intermediate or high risk prostate adenocarcinoma was terminated because “prostate cancer treatment has greatly changed since the initiation of this trial and therefore we have stopped this trial to allow for further evaluation of the treatment landscape” [26].

PSMA-targeted gold nanoparticles

Gold nanoparticles (GNPs) are another avenue of research with promising pre-clinical results, particularly when functionalized with molecules that bind to prostate-specific membrane antigen (PSMA). After intravenous injection, these PSMA-GNP nanoparticles are selectively taken up by prostate cancer tumor cells and can be readily imaged using X-ray fluorescence [27]. These nanoparticles also act as radiosensitizers, possibly through a radiation-induced bystander effect (RIBE), which makes them ideally suitable for treatment of prostate cancer metastases [28].

There are some safety concerns regarding GNPs as they degrade slowly in vivo, leading to the possibility of cumulative toxicity, particularly in the liver and spleen [29]. It has been difficult to predict toxicity of individual compounds as there are many factors involved, including the surface charge and surface chemistry of the GNPs, the size and shape of the particles, the dose and route of administration, etc. [30]. Fortunately, there are many disease areas where GNPs are being considered, which makes it important to share safety results from clinical trials [31].

#### 3.2.3. Oncolytic Virus Therapy

Oncolytic viruses specifically target cancer cells where they can replicate and infect nearby cancer cells without affecting normal tissue. Viruses in clinical development are either naturally occurring, non-pathogenic to humans, or genetically engineered [32]. Several viruses are being tested in clinical trials in men with prostate cancer [33].

#### 3.2.4. Precision Medicine

Precision medicine uses molecular testing to identify specific genetic mutations and biomarkers in an individual patient’s cancer cells that can guide treatment decisions. Examples include alterations in androgen receptor signaling (associated with resistance to bicalutamide and enzalutamide); genomic alterations leading to activation of the PTEN-PI3K-AKT pathway (increases response to capivasertib); and disruption of the DNA repair pathway (enhances response to PARP inhibitors) [34].

PSA (prostate-specific antigen) screening is a well-established method to detect prostate cancer in men. However, limitations include a high rate of false-positive results and overdiagnosis, leading to overtreatment. This has led to the development of new techniques to improve prostate cancer detection accuracy. Some examples include:Prostate Health Index (PHI)

The PHI is a blood test that measures three forms of PSA to provide a more accurate assessment of prostate cancer risk and has been shown to improve prostate cancer detection accuracy, potentially reducing the number of unnecessary biopsies in up to a third of men [35].

4Kscore Test

The 4Kscore is a blood test that measures four different PSA-related proteins to assess prostate cancer risk and can help distinguish between aggressive and non-aggressive prostate cancer, reducing the likelihood of overtreatment [36].

MRI Fusion Biopsy

An MRI fusion biopsy combines MRI images with real-time ultrasound to improve prostate cancer detection accuracy and can help to decrease the overdetection of insignificant cancers (for example, [37]).

Liquid Biopsy

A liquid biopsy is a blood test that detects circulating tumor cells (CTCs) and circulating tumor DNA (ctDNA) to help monitor prostate cancer progression and identify treatment resistance [38].

PCA3 Test

The PCA3 is a urine test that measures the expression of the prostate cancer gene 3 (PCA3) to assess prostate cancer risk and therefore reduce the number of unnecessary biopsies [39].

#### 3.2.5. Androgen Receptor Targeting

Prostate cancer is dependent on androgens for growth and progression. Several drugs are available that target androgen receptor signaling, including enzalutamide, apalutamide, and darolutamide. These drugs have been shown to improve survival in patients with metastatic prostate cancer and are being studied in men with locally advanced disease, particularly in combination with radiation therapy [40,41], although the longer term effects on changes to muscle mass and adiposity are unknown [42].

Ongoing clinical trials include ALADDIN, a randomized phase III trial of darolutamide vs. placebo [43] and APPROACH, a randomized trial of apalutamide plus androgen deprivation therapy [44].

#### 3.2.6. Pharmacological-Induced Ca^2+^ Cytotoxicity

TRPM8 (Transient receptor potential cation channel subfamily M member 8), also known as the cold and menthol receptor 1 (CMR1), is an ion channel that allows entry of Ca^2+^ ions into the cell when activated and is highly expressed in stage III/IV androgen-dependent prostate cancer. Pre-clinical studies indicate that pharmacologically induced Ca^2+^ cytotoxicity, which can be evoked by menthol, sensitizes cancer cells to standard therapies [45,46].

#### 3.2.7. Immunotherapy

A variety of immunotherapies are under development, including peptide-based cancer vaccines (for example, [47]). Although useful in cancers such as melanoma, bladder cancer, and non-small cell lung cancer, clinical studies of immunotherapy in prostate cancer have been largely disappointing [48], perhaps because of the “cold” tumor microenvironment [49]. A recent systematic review explains some of the many obstacles to be overcome [50].

#### 3.2.8. Active Surveillance

Active surveillance is the monitoring of the patient with regular PSA tests and prostate biopsies but delaying treatment until the cancer shows signs of progression. This approach is probably most appropriate for men with low-risk prostate cancer, those with a limited life expectancy (such as the elderly), or in resource-limited settings [51]. This section is included for completeness, as guidelines do not recommend active surveillance when the prostate cancer is locally advanced or confined to the lymphatics, unless treatment options are unacceptable (e.g., [52]).

## 4. Discussion

The detection and eradication of locally advanced metastatic disease is clearly an important area of research, as the spread of tumor cells to local lymphatics is a strong prognostic indicator and predictor of death in the subsequent five years. It may therefore be surprising to find only 103 studies covering this topic from 1990 to date. However, there are several technical difficulties that need to be overcome before progress can be made, including identification of small clumps of metastatic cells within the pelvis and targeting treatment to these microscopic disease areas with minimal damage to normal tissues. Recent advances in precision medicine, androgen receptor targeting, immunotherapy, radiation therapy, and active surveillance may (individually or in combination) improve therapy decisions for locally advanced prostate cancer. Note that it is important to test all treatments in adequately powered randomized clinical trials, as these provide the best evidence to inform clinical treatment guidelines. Unfortunately, the lack of such evidence means that many guidelines do not adequately take into account either the benefits or harms of treatments (for example, [53]) and instead rely on a consensus of experts [54].

Some of the shortcomings of current prostate cancer treatments include side effects of treatment (such as impotence, urinary incontinence, and bowel dysfunction); limited effectiveness of treatment; resistance to treatment (leading to disease progression); and a lack of personalized treatment tailored to individual patient characteristics, biomarkers, and genetic profiling.

Many of the studies in the review of clinical trials included a radiation intervention. The advantage of external beam radiation is that imaging is not necessary, as the treatment should target actively dividing cancer cells. However, the pelvis includes normal tissues at risk of damage from X-rays, including the lining of the gut. Several clinical trials are investigating more sensitive techniques such as PSMA PET scans [55] that can be combined with radiation therapy to the whole pelvis [56] to better target the treatment to focal areas of metastatic disease.

Future trends in prostate cancer treatment include precision medicine, immunotherapy, targeted therapy using drugs that specifically target cancer cells’ genetic and molecular characteristics, liquid biopsy to monitor treatment response and detect treatment resistance, and artificial intelligence (AI) to analyze large data sets to identify personalized treatment options. Other new treatments being developed include oncolytic virus therapy and androgen receptor targeting. One particularly exciting area of research concerns pharmacological-induced Ca^2+^ cytotoxicity, as this opens the possibility of enhancing the effectiveness of conventional treatments using a substance as simple as menthol.

One of the more promising areas of research is the use of gold nanoparticles, which are being actively investigated for imaging, diagnosis, and treatment of disease [57]. When linked to PSMA-directed antibodies, the gold nanoparticles only accumulate around prostate cancer cells and, by taking advantage of radiation-induced bystander effects, enable the use of relatively low doses of X-rays to effectively destroy the metastatic disease with little or no harm to surrounding normal tissues. 

Gold nanoparticles have additional advantages. They are non-toxic and can be produced in an environmentally friendly manner, potentially from acai berry or elderberry extracts [58] or directly from aquatic plants grown in aqueous gold solutions [59]. From a global health perspective, another potential advantage is that gold nanoparticles can be used as radiosensitizers with widely available conventional external beam X-ray radiation therapy [60], including kilovoltage X-rays that are ideally suited to a limited-resource setting [61]. This is particularly relevant in geographic areas where the highest prostate cancer mortality rates are found, such as the Caribbean, sub-Saharan Africa, and parts of the former Soviet Union [62]. Unfortunately, very few clinical trials have been carried out in low- and lower-middle-income countries [63].

Men recently diagnosed with locally advanced or advanced prostate cancer have many concerns about symptoms, treatment options, and the side-effects of treatments. Fundamentally, they want to have answers regarding their length and quality of life. While there is good advice available for coping with tiredness, pain, bladder problems, eating difficulties, etc., it is not currently possible to “see into the future” and predict what life will be like one year (or even one month) from now. However, examples from other areas of medicine have demonstrated that research produces therapies that can transform a disease from a dreaded killer to an annoying nuisance in a very short space of time. It is therefore worth maintaining a sense of optimism in the face of adversity, not only for oneself but also for the brothers, sons, nephews, and others who may face such a diagnosis in the future. It is this hope that drives the basic research and clinical trials that will provide evidence of the benefits and harms of new treatments, and eventually consign prostate cancer to a footnote in history alongside bubonic plague, chronic myelogenous leukemia, and COVID-19.

## 5. Conclusions

This article has analyzed clinical trials and reviewed some of the recent advances in therapy decisions for locally advanced prostate cancer and shown that more clinical trials are needed, particularly in countries outside of North America and Europe. There are potentially many promising new technologies that need to be translated from bench to bedside to inform clinical guidelines and reduce global mortality from prostate cancer.

## Figures and Tables

**Figure 1 jpm-13-00938-f001:**
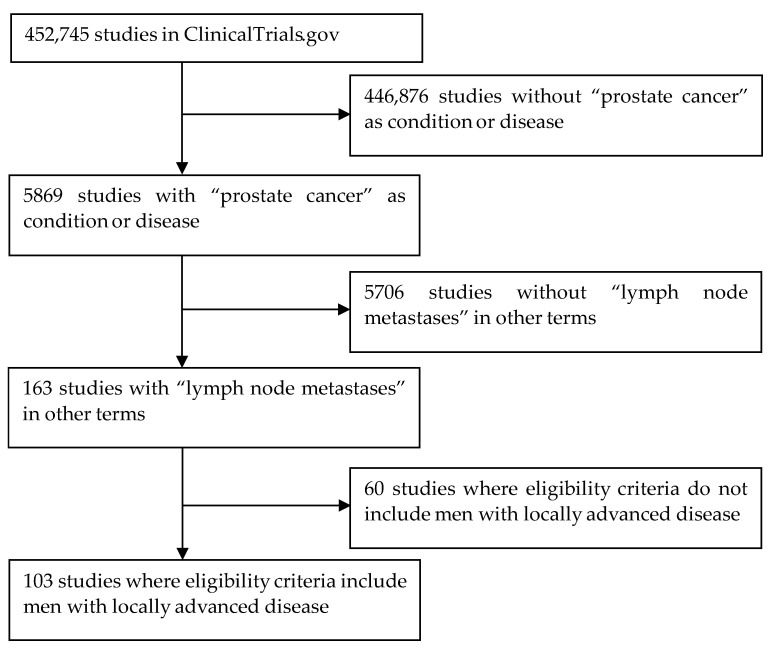
Selection of 103 studies from ClinicalTrials.gov.

**Figure 2 jpm-13-00938-f002:**
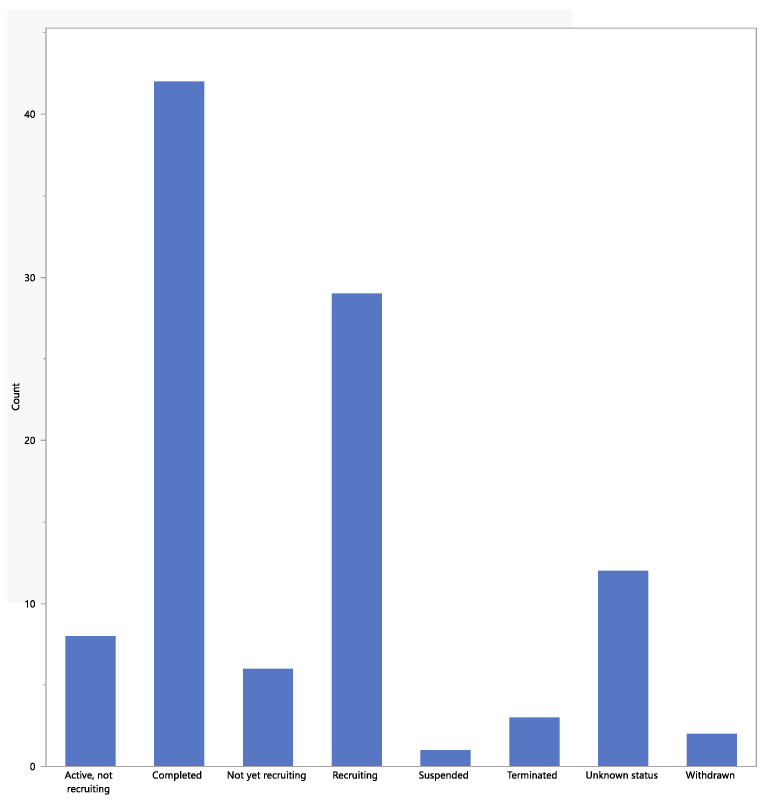
Status of the studies.

**Figure 3 jpm-13-00938-f003:**
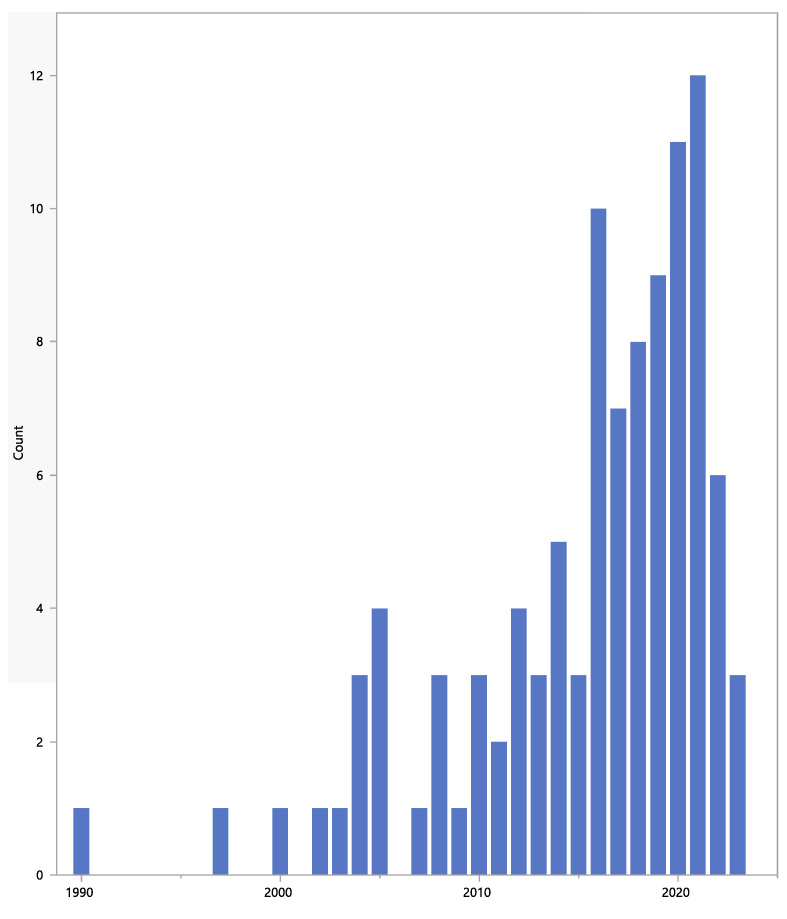
Start dates.

**Figure 4 jpm-13-00938-f004:**
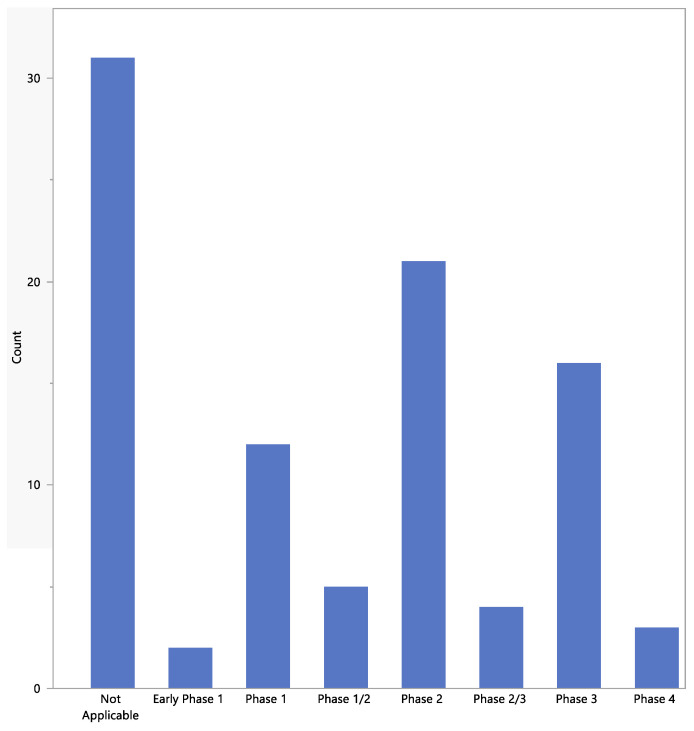
Phases of the studies.

**Figure 5 jpm-13-00938-f005:**
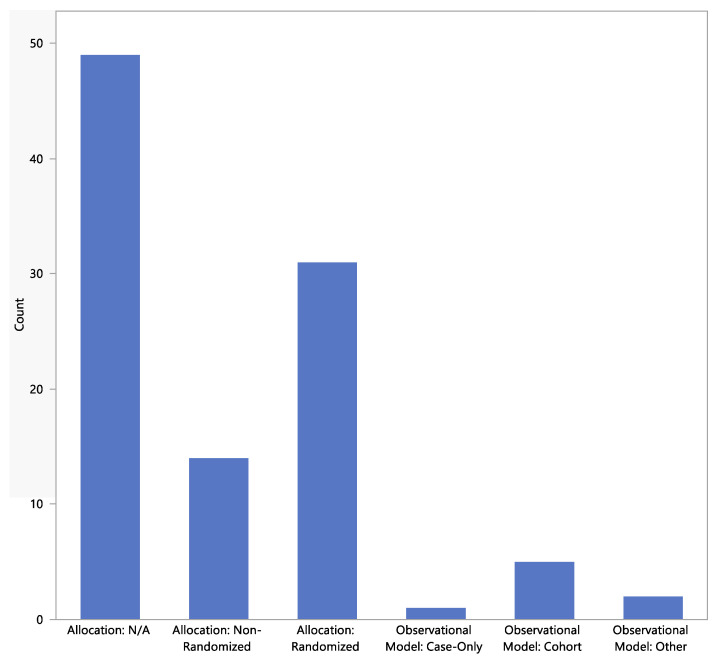
Study designs (treatment allocation).

**Figure 6 jpm-13-00938-f006:**
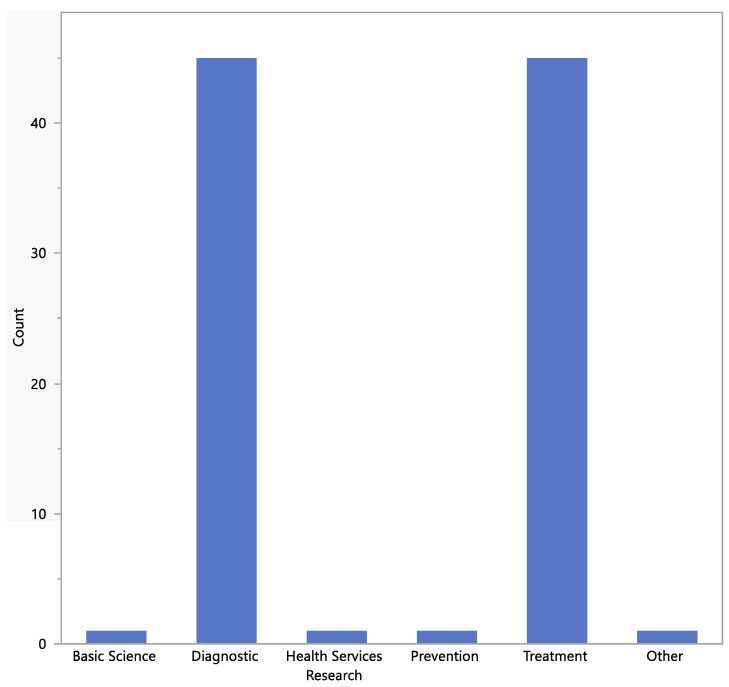
Study designs (purpose of study).

**Table 1 jpm-13-00938-t001:** Geographic distribution of country of PI.

Country of PI	Number of Studies
USA	36
Canada	10
Colombia	1
Brazil	1
China	7
Denmark	6
France	6
Italy	6
Netherlands	6
Belgium	5
Switzerland	4
Germany	3
Norway	2
Sweden	2
Austria	1
Finland	1
Poland	1
UK	1
Turkey	1
Israel	1
Japan	1
Korea	1

**Table 2 jpm-13-00938-t002:** Studies with a planned start date in 2022 or 2023 (grouped by study design, sorted by status).

NCT Number	Status *	Planned Enrollment	Study Design
NCT05613023	Recruiting	536	Randomized: curative radiotherapy to the prostate and lymph glands to that of prostate alone
NCT05116475	NYR	152	Randomized: Darolutamide vs. placebo
NCT04655365	Recruiting	50	Non-randomized: 18 F-DCFPyL-PSMA PET/CT
NCT05269550	Recruiting	50	Non-randomized: evaluation of imaging biomarkers post radiotherapy
NCT05754580	NYR	53	Non-randomized: High-dose rate (HDR) brachytherapy boost with stereostatic body radiation therapy (SBRT)
NCT05847166	NYR	80	Non-randomized: [99 mTc]Tc-PSMA-T4
NCT05596851	Recruiting	15	Observational: β-probe and 68 Ga-PSMA-11 PET/CT
NCT05252806	NYR	30	Observational: Quantitative mpMRI to predict metastatic potential of prostate cancer
*NCT05705700*	*Withdrawn*	*0*	

* NYR: Not yet recruiting.

**Table 3 jpm-13-00938-t003:** Interventions used in the studies.

Intervention	Number of Studies
Drug	45
Procedure	27
Radiation	22 *
Diagnostic test	14
Biological	5
Device	6
Other	10
Not applicable (observational study)	3

NOTE: Some studies have several interventions (e.g., drug plus radiation). * See Table 4 for details.

**Table 4 jpm-13-00938-t004:** Types of radiation intervention.

Type of Radiation Intervention	Number of Studies
Radiation therapy	9
Radiation to the prostate bed with or without addition of lymph node irradiation	1
Hypofractionation	2
Intensity-modulated radiation therapy	2
Intensity-modulated radiotherapy of the pelvic lymph nodes using simultaneous integrated boost (SIB)	1
Stereotactic ablative radiotherapy	1
Simultaneous integrated boost (SIB) dose-escalation radiotherapy	1
Stereotactic body radiotherapy (SBRT)	2
Stereotactic ablative radiotherapy	1
Brachytherapy	1
High-dose brachytherapy	1
Proton plus carbon ion radiation	1
[99 mTc]Tc-PSMA-T4	1
PET scan, whole-body MRI	1

NOTE: Some studies have more than one radiation intervention.

## Data Availability

All data used in this work can be found in Appendix A.

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
