# Peer review of "Analysis of Clinical Trials and Review of Recent Advances in Therapy Decisions for Locally Advanced Prostate Cancer"

_jpm, 2023, doi:10.3390/jpm13060938_

Round 1

Reviewer 1 Report

The manuscript entitled “Recent Advances in Therapy Decisions for Locally Advanced Prostate Cancer”, is a review paper that summarizes the bibliography of recent clinical trials and other studies regarding potential therapeutical approaches against locally advanced prostate cancer. This study is particularly meticulous and the results obtained are really interesting. It is well written and very didactic and I do not find any significant incorrectness.

Nevertheless, the following issues need to be addressed:

- The resolution (quality) of graphics and images could be improved

- The author should also mention in the results or (better) discussion published non-clinical but very promising studies. For instance the study DOI: 10.1038/s41419-020-03256-5, regarding drug targeting of locally advanced prostate cancer, was obtained through the collaboration of various academic groups and hospitals in Italy and Switzerland. Similarly, the publication DOI: 10.1016/j.redox.2017.10.009 (although not concerning locally advanced PCa) was achieved through the efforts of several groups from Turkey, Germany, and Hungary.  Perhaps it would be appropriate for the author to include these references.

Author Response

The author thanks the reviewer for their kind words and useful suggestions. A revised manuscript highlighting additions made in response to all reviewers has been uploaded, so that the reviewers can see the implementation of their suggestions in context. Specifically:

  • Higher resolution figures have been prepared and will be uploaded as a ZIP file, in accordance with a suggestion from the editor.
  • A new section has been added with references [3.2.6 Pharmacological-induced Ca2+ cytotoxicity] see lines 237 to 243, and mentioned in the Discussion at lines 296 to 298.

Reviewer 2 Report

Congratulation to the author for the idea of reviewing the topic of new treatments in locally advance prostate cancer. However, the paper is a bit confusing in terms of objetive, methodology and information provided; 

-        Methodology about the literature review should be further explained. How were the papers specifically selected? Are all locally advance or those with Lymph nodes? It is not clear to the reader. Furthermore, the new treatments information (i.e precision medicine, immunotherapy…) is too general and not sure focus in this specific disease stage.

-        The paper looks more a review about current research in advance prostate cancer, but it does not give results to be analyse. The title should be changed accordingly.

-        The information about Gadolinium-based nanoparticles  and Hafnium oxide particles seems no to include prostate studies. Why the authors include them? If the researches are not focus on prostate cancer, they should be discarded.

-        Why the authors include active surveillance as an option? The information is too superficial without a clear focus. The author should better define his objective.

Author Response

The author thanks the reviewer for their useful suggestions. A revised manuscript highlighting additions made in response to all reviewers has been uploaded, so that the reviewers can see the implementation of their suggestions in context. Specifically:

  • More explanation of the literature review has been added at lines 52-54 and lines 130-132.
  • The title has been changed to better reflect the focus of the article (see lines 2-3).
  • Prostate cancer specific information about Gadolinium-based nanoparticles  and Hafnium oxide particles has been added at lines 163-164 and 169-170.
  • An explanation about the role of active surveillance has been added at lines 258-260.

Reviewer 3 Report

In the manuscript, the authors analyzed relevant clinical studies listed on ClinicalTrials.gov, combined with a short literature review that considers new therapeutic approaches that can be investigated in future clinical trials. The manuscript couldn’t be accepted until following questions be addressed.

1.       The introduction should explain why prostate cancer clinical studies in the ClinicalTrials.gov database are analyzed.

2.       Although gold nanoparticals could be used as radiotherapy sensitizer for prostate cancer treatment, it is difficult to degrade in vivo and has the possibility of cumulative toxicity. The authors should discuss its biological safety.

3.       The results of immunotherapy for prostate cancer are generally unsatisfactory, mainly due to the effect of dense prostate cancer envelope, and the authors should add discussion about this.

4.       In Section 3.2.4, the authors should add examples of PSA screening and related applications of new prostate cancer screening methods.

5.       The authors should add to the discussion of the shortcomings of current prostate cancer treatment and future trends.

The quality of English Language should be improved.

Author Response

The author thanks the reviewer for their useful suggestions. A revised manuscript highlighting additions made in response to all reviewers has been uploaded, so that the reviewers can see the implementation of their suggestions in context. Specifically:

  • An explanation of ClinicalTrials.gov has been added to the Introduction at lines 39-41.
  • Safety concerns of gold nanoparticles have been metioned in lines 179-185.
  • The unsatisfactory results of immunotherapy in prostate cancer have been discussed in lines 247-251.
  • Section 3.2.4 has been expanded as suggested (lines 200-224).
  • Shortcomings (lines 279-283) and future trends (lines 291-298) have been added to the Discussion.

Round 2

Reviewer 1 Report

The author has addressed all reviewers' comments/suggestions. I found their responses entirely satisfactory and the revised version has been much improved. I now recommend the paper for publication in JPM

Author Response

Thank you.

Reviewer 2 Report

The revision paper does not answer any of the comments provided.

If AS is not a correct management for this stage the authors should eliminate this section.

There is still lack of information about how they performed the research search, which terms and so on....

The comment about the infomation about gadolinium-based nanoparticles and hafnium should summarize the research and results available in prostate and analyse if there is currently any trial in this pathology specfically? are these approaches develop for this stage of the disease? How the authors find this information specifically in locally advance disease?

furthermore, goind deeper in the clinical trials selected;

this study is for low risk PCa why and how the authors include it in the manuscript?

NCT05343936

Recruiting

200

Observational: Evaluation of an active surveillance program for prostate cancer in Brazil

Author Response

Thank you for your additional comments. A copy of the revised manuscript with changes highlighted in yellow has been provided. The changes are summarised below.

The article is intended to be an overview of "what is here, and what might be coming", without including unfounded speculation. This is important because the field is rapidly changing and there are many promising areas of research and even specialists find it difficult to keep up to date. Text describing the intention of this article has been added: "This information will be useful to a man with a first diagnosis of locally advanced prostate cancer who has not yet undergone surgical or radiation therapy and wishes to know the current “state of the art” for men in this situation." (lines 44 - 48)

To reply to each comment in turn:

"If AS is not a correct management for this stage the authors should eliminate this section."

Active Surveillance is often discussed with respect to prostate cancer, but as the reviewer has stated this should not be offered to men with locally advanced disease (unless there are exceptional circumstances such as much reduced life expectancy). To remove this section entirely might lead the reader to assume that AS is a reasonable option. "This section is included for completeness" has been added (line 303).

"There is still lack of information about how they performed the research search, which terms and so on...."

The literature search was not carried out systematically, as it proved to be very difficult to obtain relevant information. For example, a PubMed search of "hafnium AND prostate" finds only one reference (and it is not relevant). Most of the information was obtained by carefully reading review articles and going to the cited sources. This was supplemented with searches on manufacturers' websites. The following has been added to the Materials and Methods section: "The literature review was not carried out systematically. Most of the information was obtained from review articles and their cited sources. In addition, manufacturers websites were scanned to determine if there were clinical trials in the pipeline." (lines 57 - 59).

"The comment about the infomation about gadolinium-based nanoparticles and hafnium should summarize the research and results available in prostate and analyse if there is currently any trial in this pathology specfically? are these approaches develop for this stage of the disease? How the authors find this information specifically in locally advance disease?"

For the gadolinium-based nanoparticles, no studies have yet been carried out in locally advance prostate cancer, and the following text has been added: "To date, there is only pre-clinical evidence from human prostate cancer cell lines (DU145 and PC3) to support a possible role as a radiosensitizer in prostate cancer [22], therefore there is a potential for the use of this compound in a clinical trial of men with locally advanced prostate cancer." (lines 201 - 204).

For hafnium oxide particles, there is pre-clinical evidence and one clinical study that was terminated. The following text has been added: "No clinical trials of NBTXR3 have completed in prostate cancer, although pre-clinical evidence in mice and patient derived xenograft models is encouraging [25]. A phase 1/2 study of NBTXR3 nanoparticles and radiation therapy in the treatment of intermediate or high risk prostate adenocarcinoma was terminated because “prostate cancer treatment has greatly changed since the initiation of this trial and therefore we have stopped this trial to allow for further evaluation of the treatment landscape” [26]." (lines 209 - 215).

Both gadolinium and hafnium particles are being researched intensively in other cancers, but there are as yet no ongoing clinical studies in men with locally advanced prostate cancer. This is an exciting and rapidly evolving area of research.

"furthermore, goind deeper in the clinical trials selected;..."

Based on this comment, a fresh data set was obtained and re-analysed. Figure 1 is now a flow diagram systematically showing the steps required to obtain the 103 studies analysed in the article. The author thanks the reviewer for pointing out this flaw which has now been rectified.

Reviewer 3 Report

The manuscript could be accepted now.

Author Response

Thank you.